

# Unified system describing factors related to the eradication of an alien plant species

Halina Galera, Agnieszka Rudak and Maciej Wódkiewicz

Faculty of Biology, University of Warsaw, Warsaw, Poland

## ABSTRACT

**Background**. In the field of biological invasions science, a problem of many overlapping terms arose among eradication assessment frameworks. Additionally there is a need to construct a universally applicable eradication evaluation system. To unify the terminology and propose an eradication feasibility assessment scale we created the Unified System for assessing Eradication Feasibility (USEF) as a complex tool of factors for the analysis of eradications of alien (both invasive and candidate) plant species. It compiles 24 factors related to eradication success probability reported earlier in the literature and arranges them in a hierarchical system (context/group/factor/component) with a possibility to score their influence on eradication success.

**Methodology**. After a literature survey we analyzed, rearranged and defined each factor giving it an intuitive name along with the list of its synonyms and similar and/or related terms from the literature. Each factor influencing eradication feasibility is ascribed into one of four groups depending on the context that best matches the factor: location context (size and location of infestation, ease of access), species context (fitness and fecundity, detectability), human context (knowledge, cognition and resources to act) and reinvasion context (invasion pathways). We also devised a simple ordinal scale to assess each factor's influence on eradication feasibility.

**Conclusions**. The system may be used to report and analyze eradication campaign data in order to (i) prioritize alien species for eradication, (ii) create the strategy for controlling invasive plants, (iii) compare efficiency of different eradication actions, (iv) find gaps in knowledge disabling a sound eradication campaign assessment. The main advantage of using our system is unification of reporting eradication experience data used by researchers performing different eradication actions in different systems.

## INTRODUCTION

The outcome of biological invasion management and combat depends on many environmental, site and species specific factors. Each of these factors varies in a separate way and may affect the course of an eradication action in a different manner. For this reason, it is very difficult to predict the course of these actions and analyze the reasons for their success or failure. Papers describing factors associated with eradication success and their evaluation (for summaries see *e.g.*, *Kettenring & Adams, 2011*; *Pluess et al., 2012a*; *Pluess*

Corresponding author
Agnieszka Rudak,
a.rudak@biol.uw.edu.pl

*et al., 2012b*; *Brunel, Brundu & Fried, 2013*; *Dodd et al., 2015*; *Panetta & Gooden, 2017*) accumulate; however, they lack comparability on a global scale. The described systems are mostly local, prepared in accordance with country-specific legislation and management scenarios. This facilitates comparison of eradication actions performed within country limits (*i.e.*, Australia) but makes the comparison of eradications performed in different parts of the world a demanding task. This is true not only because of delimitation of different factors, but also because the used factor evaluation scales differ. Several milestone papers summarizing current biological invasion knowledge (*e.g.*, *Pyšek, 1995*; *Richardson et al., 2000*; *Genovesi, 2001*; *Colautti & MacIsaac, 2004*; *Blackburn et al., 2011*; *Blackburn et al., 2014*; *Robertson et al., 2020*) stressed the necessity to unify the description and evaluation of eradication related factors. Other papers distinguish further factors as terms (*e.g.*, *Panetta et al., 2011*; *Pluess et al., 2012a*; *Dodd et al., 2015*), presented in a form of questionnaire (*e.g.*, "Can the species reproduce through vegetative fragmentation?" in *Panetta & Timmins, 2004*, for further examples see *Cunningham et al., 2004*; *Panetta, 2015*; *Blood et al., 2019*) or affirmative sentences (*e.g.*, "Project leaders must be energetic, optimistic, and persistent in the face of occasional setbacks" in *Simberloff, 2009*; "the biology of the species must be appropriate" in *Simberloff, 2013*). Even though the effort to unify invasion management terminology has been made (*Robertson et al., 2020*), at this time no paper summarizing the eradication related factors has been published.

Our work is predominantly focused on plant invasions and was inspired by *Dodd's et al. (2015)* article "Plant extirpation at the site scale: implications for eradication programmes". We analyzed the eradication factors reported in the literature (*e.g.*, *Rejmánek & Pitcairn, 2002*; *Skurka Darin et al., 2011*; *Pluess et al., 2012a*; *Dodd et al., 2015*; *Dana et al., 2019*; *Williams et al., 2019*) supplementing them and adding a more detailed description based on available literature and our own experience gained during the eradication campaign of annual bluegrass (*Poa annua* L.) in the Antarctic (*Galera et al., 2017*; *Galera et al., 2018*; *Galera et al., 2019*; *Galera et al., 2021*). We arranged these factors in a new way constructing a Unified System for assessing Eradication Feasibility (USEF). Our system mainly concerns the process of eradication defined as "complete and permanent removal of all wild populations of a species from a defined area by means of a time-limited campaign" (*Genovesi, 2001*; *Genovesi, 2005*; *Brunel, Brundu & Fried, 2013*). Due to its open nature it may also be useful for extirpation and control campaigns.

The aim of our paper is to clarify the used terminology at the present stage of research on eradication feasibility assessment. Our goal was to construct the most intuitive nomenclature system. We also included synonyms as well as similar and related terms reported in the literature. When constructing the definitions, we tried to use already existing terms by quoting them directly. Papers on this topic are still being published at a fast pace, so the time is ripe for such a synthesis before we experience "information buzz" in the field of eradication due to overload of data presented in varying ways. We also attempted to construct an evaluation scoring system based on simple and possible to define categories to assist eradication action feasibility assessment and comparison.
## SURVEY METHODOLOGY

We conducted a comprehensive literature search including both original papers and reviews. In our search we used the following keywords: eradication feasibility/success/assessment/framework to look for papers published in any time period. We identified c.a. 500 articles connected to our field of interest, most of them found through Google Scholar search engine. Other sources were Web of Science (Ecology and Biodiversity Conservation category) and JSTOR database (Ecology and Evolutionary Biology category). After abstract screening for data relevant to eradication assessment, and later studying the relevant papers, we reduced the number of publications to 37 articles that included different usage of terms related to eradication research and various types of comprehensive frameworks for eradication assessments. These articles were then used to create the USEF system and are cited in Tables 1 and 2 next to the relevant factors.

### System design

Based on our literature review we selected eradication related factors and ascribed them into four groups depending on the eradication context that matches the factors best (Fig. 1). The four contexts represent the four main aspects associated with the eradication of an alien species. The factors are further organized into groups combining factors of a similar nature. The rationale was to organize the factors in a way that facilitates work with the system. The first group, "location context" is therefore related to the spatial characteristics of the infestation and considers its size and location (factors 1–5) as well as ease of access for the necessary actions to be performed (factors 6 and 7). The next group is the "species context" which groups factors related to the biology of the considered target (invasive) species. Factors here are related to fitness and fecundity (factors 8–12) and detectability of the species (factors 13–14). In the next group, the "human context", factors related to the human (personnel, managers, organizations) behavior influencing the eradication campaign are grouped. The included factor groups are knowledge (factors 15–17) and cognition and resources necessary for alien plant management (factors 18–23). The last context is the "reinvasion context", which contains only one factor, 'invasion pathways' (factor 24).

The complex nature of the selected factors raised problems with classifying some of them into one group as they are relevant for at least two groups (*e.g.*, factor 5 'monitoring rate' and factor 8 'adaptation to new climate conditions' depend on location and biology of managed population—they could be classified into both location and species context). In such instances we fitted the factors into groups that match the factor better. Regardless of the assignment of factors to different groups they should be considered jointly, as they all interact in shaping the success or failure of a specific eradication.

### Selection of factors for the system

We considered 22 factors mentioned by *Dodd et al. (2015)*. Some of them were redefined and expanded to improve their universal usage. In several instances it was necessary to further subdivide the factors into several components, which may counteract with each other and have a contrasting effect on eradication success (Table 1). Additionally we

**Table 1  Description of factors affecting the probability of eradication of an alien plant species included in the Unified System for assessing Eradication Feasibility (USEF).**

| Group | Factor | Definition and factor components | Synonyms or similar and related terms |
|---|---|---|---|
| Size and location | 1. Infestation size | The area to which treatment is applied together with information on specific eradication stage. | *Similar terms:* net infestation size (*Rejmánek & Pitcairn, 2002*; *Panetta & Timmins, 2004*), net invaded area (*Corbin et al., 2017*), area infested (*Pluess et al., 2012a*), size of infested area (*Pluess et al., 2012b*), area currently infested by the weed (*Cunningham et al., 2004*), area of infestation newly detected (*Panetta & Lawes, 2007*) <br> *Related term:* the size of the invasion—adult plants per hectare (*Cacho et al., 2006*) |
| | 2. Number of separate infestation sites | Number of discrete infected areas together with information on specific eradication stage. | *Synonyms:* number of separate sites (*Dodd et al., 2015*), number of infestations (*Cunningham et al., 2004*; *Panetta, 2015*), number of separate infestations (*Rejmánek & Pitcairn, 2002*), number of discrete infestations (*Panetta & Timmins, 2004*) <br> *Similar term:* pest distribution (*Pluess et al., 2012a*) |
| | 3. Isolation of infestation | Degree of spatial and/or ecological isolation of the target population. | *Similar term:* insularity (*Pluess et al., 2012a*; *Pluess et al., 2012b*) |
| | 4. Monitoring area size | Infestation size (total area ever infested, *Burgman et al., 2013*) plus the surrounding area that must be searched in return trips following treatments (slightly changed definition of gross infestation size by *Blood et al., 2019*). | *Synonym:* total area ever infested (*Burgman et al., 2013*) <br> *Similar terms:* gross infestation size (*Panetta & Timmins, 2004*), gross invaded area, total search area (*Corbin et al., 2017*), total gross area (*Panetta, 2015*), gross area of the weed infestation (*Blood & James, 2018*; *Blood et al., 2019*); total invaded area (*Panetta & Lawes, 2007*), total infested area (*Panetta & Lawes, 2007*; *Panetta et al., 2011*), total area currently infested (*Burgman et al., 2013*) <br> *Related terms:* initial gross infestation area (*Rejmánek & Pitcairn, 2002*); proportion infested [%] (*Pluess et al., 2012a*); density of mature plants per ha (*Corbin et al., 2017*); occupied zone, buffer zone (*Fletcher et al., 2015*) |
| | 5. Monitoring rate (*Dodd et al., 2015*; *Dodd et al., 2017*) | The monitoring intensity, which includes: <br> (a) duration of annual monitoring period (duration of vegetative season), <br> (b) recommended monitoring frequency per annual monitoring period. | *Related terms:* length of monitoring after all infestations have been cleared (*Howell, 2012*), number of follow-up visits required (*Skurka Darin et al., 2011*), (b) search frequency (*Blood et al., 2019*), frequency of post-treatment reviews (*Dana et al., 2019*), mean annual visitation rate, site visitation frequency (*Panetta, 2007*), minimum surveillance rate per annum (*Dodd et al., 2017*) |

**Table 1** (*continued*)

| Group | Factor | Definition and factor components | Synonyms or similar and related terms |
|---|---|---|---|
| **Ease of access** | 6. Land use and ownership of infested area | In the context of predicting the success of eradication, land use and land management practices include: (a) type of land management practices (type of human activity involved in land use), (b) accessibility resulting from land ownership relations, (c) complexity of a management mosaic (*Epanchin-Niell et al., 2010*)—land ownership mosaic and the resulting heterogeneity of land use. | *Similar term:* land use (*Panetta & Timmins, 2004*; *Dodd et al., 2015*) *Related terms:* accessibility (*Pluess et al., 2012a*), number of separate land managers (*Dodd et al., 2015*), landowner cooperation (*Skurka Darin et al., 2011*), adequate lines of authority (*Simberloff, 2013*); "Are the necessary permits for the control treatment expected to be obtained?" (*Corbin et al., 2017*), "the action plan is legal and meets all administrative requirements (permits of landowner, authorisations of responsible institutions, authorization to use chemical compound, etc.)" (*Dana et al., 2019*), permission of access from property owners (*Gardener, Atkinson & Rentería, 2010*) (c) "Within the invaded area, do all the agencies, organizations, and/or landowners agree to participate?" (*Corbin et al., 2017*) |
| | 7. Accessibility | Ease of access to the infested area by personnel performing eradication action. This includes: (a) distance to nearest eradication management office, (b) difficulties in gaining access to infested area due to landform (*Williams et al., 2019*), ability to move through vegetation, travel time (*Blood et al., 2019*) and flooding (*Panetta & Timmins, 2004*; *Blood et al., 2019*), except difficulties described in factor 6, points b and c. | *Synonyms:* accessibility of infestation site (*Skurka Darin et al., 2011*) (a) general accessibility (*Dodd et al., 2015*) (b) ease of access (*Cunningham et al., 2004*), general accessibility of infestations (*Panetta & Timmins, 2004*), site accessibility (*Blood et al., 2019*), "the action plan area is entirely accessible for workers" (*Dana et al., 2019*) *Related term:* driving time to site (*Skurka Darin et al., 2011*) |
| **Fitness and fecundity** | 8. Adaptation to new climate conditions | The degree of adaptation of the target population to new climatic conditions. This includes: (a) adaptive capabilities of the invasive species, (b) similarity of climatic conditions in the continuous (native or non-native) range of the species | *Similar terms:* climate match of species with site (*Dodd et al., 2015*), similarities among the climates of the invaded area and endemic population (*Zamora, Thill & Eplee, 1989*) *Related terms:* indoor or outdoor habitat (*Pluess et al., 2012a*), climatic suitability (*Kaplan et al., 2014*) |
| | 9. Number and distribution of propagules | The size and spatial structure of the propagule store in the infested community. This includes: (a) seed bank size/density (*Kaplan et al., 2014*; *Williams et al., 2019*), (b) propagule distribution (*Williams et al., 2019*)—horizontal and vertical distribution of propagules in soil and on its surface. | *Similar terms:* number of propagules (*Dodd et al., 2015*), reproductive output (*Kaplan et al., 2014*) (b) spatial structure of the seed bank (*Wódkiewicz et al., 2014*), distribution of the seed bank (*Williams et al., 2019*) *Related terms:* presence of resistance structures, *e.g.*, seed bank, spores, cysts (*Dana et al., 2019*); dispersal mechanisms (*Williams et al., 2019*), propagule dispersal (*Blood et al., 2019*), mode of propagule dispersal (*Panetta & Timmins, 2004*); germination rate, juvenile survival (*Cacho et al., 2006*) |

**Table 1** (*continued*)

| Group | Factor | Definition and factor components | Synonyms or similar and related terms |
|---|---|---|---|
| | 10. Vegetative propagation | The ability of the plant to produce vegetative propagules. | *Synonym:* vegetative reproduction (*Skurka Darin et al., 2011*; *Dodd et al., 2015*; *Blood et al., 2019*) <br> *Similar term:* vegetative fragmentation (*Panetta & Timmins, 2004*) |
| | 11. Propagule longevity (*Panetta & Timmins, 2004*; *Dodd et al., 2017*) | Maximum longevity of seeds or vegetative propagules (*Panetta & Timmins, 2004*; *Panetta & Cacho, 2014*). | *Similar terms:* maximum longevity of seeds or detachable vegetative propagules (*Blood & James, 2018*; *Blood et al., 2019*), seedbank or propagule longevity (*Skurka Darin et al., 2011*), seed bank duration, maximum longevity of seeds (*Corbin et al., 2017*), seedbank longevity (*Cunningham et al., 2004*; *Fletcher et al., 2015*; *Neville, Fujikawa & Halabisky, 2019*), seed bank lifetime (*Fletcher et al., 2015*), seed persistence (*Panetta, 2015*), maximum seed longevity (*Panetta & Lawes, 2007*), maximum longevity of soil-stored seed/of the soil seed bank (*Burgman et al., 2013*) |
| | 12. Pre-reproductive period (*e.g.*, *Panetta & Timmins, 2004*; *Panetta, 2007*; *Panetta, 2016*; *Dodd et al., 2015*; *Corbin et al., 2017*) | Minimum length of the pre-reproductive period (*Panetta & Timmins, 2004*; *Panetta & Cacho, 2014*)—the time between seedling emergence and propagule production by the target species. | *Synonyms:* time to maturity (*Cacho et al., 2006*), age to maturation (*Corbin et al., 2017*), length of the pre-reproductive period (*Panetta & Timmins, 2004*; *Panetta & Cacho, 2014*), length of juvenile phase (*Skurka Darin et al., 2011*), juvenile period (*Panetta, 2015*; *Panetta, 2016*; *Blood & James, 2018*; *Blood et al., 2019*) |
| **Detectability** | 13. Detection possibility | Possibility of target species detection can be described as: <br> (a) species search distance (*Dodd et al., 2015*)—the distance at which plants can be detected when searching (*Williams et al., 2019*) by sight; <br> (b) possibility of detection of the target species by other methods, *e.g.,* by weed eradication detector dogs (*Cherry et al., 2016*) or using remote sensing; <br> (c) possibility to distinguish target species from other organisms. | *Similar terms:* detectability (*Panetta & Lawes, 2007*; *Skurka Darin et al., 2011*; *Burgman et al., 2013*; *Kaplan et al., 2014*; *Dodd et al., 2017*); technologies available for search—local/passive/remote detection probability (*Spring & Cacho, 2015*); ability to detect the target plant, "Will the invasive plant always be so difficult to find in the surrounding vegetation that there is a risk of project failure?" (*Corbin et al., 2017*) <br> (a) detectability [in meters] (*Panetta & Cacho, 2014*), search/detection distance (*Blood & James, 2018*; *Blood et al., 2019*) <br> *Related terms:* identification method ("how easily the organism can be identified", *Pluess et al., 2012a*), "For plants that reproduce by propagules, how detectable is the species prior to reproduction?" (*Panetta & Timmins, 2004*) <br> *Related terms:* hidden or hibernating individuals (*Dana et al., 2019*) <br> (c) "Is the species conspicuous within the matrix of invaded vegetation?" (*Cunningham et al., 2004*) |

**Table 1** (*continued*)

| Group | Factor | Definition and factor components | Synonyms or similar and related terms |
|---|---|---|---|
| | 14. Annual period of detectability prior to seed set (*Dodd et al., 2015*) | Annual period during which the species is detectable (*i.e.*, has above ground parts) prior to seed set (*Dodd et al., 2015*) | *Similar term:* detectability period (*Blood & James, 2018*; *Blood et al., 2019*) |
| **Knowledge** | 15. Knowledge of current location of infestation sites | Availability of information about current location of all infected sites. | *Synonym:* knowledge of current locations (*Dodd et al., 2015*), delimitation criterion (*Panetta & Lawes, 2005*; *Gardener, Atkinson & Rentería, 2010*) |
| | 16. Understanding of species biology (*Dodd et al., 2015*) | Knowledge of invasive population biology to eradicate the infestation. | *Synonyms:* understanding of population biology (*Gardener, Atkinson & Rentería, 2010*), knowledge and preparedness to act (*Pluess et al., 2012a*; *Pluess et al., 2012b*), "the target species must be studied well enough to suggest vulnerabilities" (*Simberloff, 2009*), "enough must be known about the biology of the target species" (*Simberloff, 2013*) |
| | 17. Eradication achieved elsewhere (*Dodd et al., 2015*) | Availability of experience gained during successful eradication of other infestations of target species. | *Synonym:* [eradication] outcome (*Pluess et al., 2012a*) *Related terms:* knowledge of treatment history at the site (*Skurka Darin et al., 2011*), prior invasion history (*i.e.*, invasive elsewhere) (*Sohrabi et al., 2020*) |
| **Cognition and resources** | 18. Reaction time (*Pluess et al., 2012a*; *Pluess et al., 2012b*) | The time elapsing between the arrival (or detection) of the organism and the start of the eradication campaign (*Pluess et al., 2012a*). | *Related term:* residence time (*e.g., Panetta, 2016*; *Blood & James, 2018*) |
| | 19. Applicable control methods | Available control measures. This includes: (a) physical control—uprooting, burning, chipping and other methods of plant material disposal (*Pluess et al., 2012a*, slightly changed); (b) cultural control—changed crop rotation, planting of resistant hosts (*Pluess et al., 2012a*); (c) chemical control (*Pluess et al., 2012a*); (d) biological control—biocontrol measures, including Sterile Insect Techniques (*Pluess et al., 2012a*) and bioinsecticides. | *Related terms:* available control measure (*Dodd et al., 2015*); methodology effectiveness, efficiency and impact (*Dana et al., 2019*); surveillance methods, technologies available for treatment (*Spring & Cacho, 2015*) |
| | 20. Personnel awareness | The level of knowledge and sense of responsibility of personnel involved in the campaign. | *Similar terms:* sufficient enthusiasm of project leaders (*Dodd et al., 2015*), "project leaders must be energetic, optimistic, and persistent in the face of occasional setbacks" (*Simberloff, 2009*), availability of specialized staff (*Dana et al., 2019*), *Related term:* biological knowledge and preparedness to act (*Pluess et al., 2012a*; *Pluess et al., 2012b*); needs of—weed researchers, botanists, farmers, land managers (*Sohrabi et al., 2020*) |

**Table 1** (*continued*)

| Group | Factor | Definition and factor components | Synonyms or similar and related terms |
|---|---|---|---|
| | 21. Coordination between monitoring agencies | Degree of cooperation between all parties involved in the eradication campaign. | *Synonyms:* coordination (*Pluess et al., 2012a*), consensus of involved administrations/departments (*Dana et al., 2019*) *Related terms:* "existence of a person or agency with the authority to enforce cooperation" (*Simberloff, 2009*), "single agency must be responsible for eradication" (*Dodd et al., 2015*), needs of government agencies (*Sohrabi et al., 2020*) |
| | 22. Sufficient allocation of resources (*Dodd et al., 2015*) | Sufficient resources allocated at the start to finish the project, including post-eradication surveys and follow-up, if necessary (*Simberloff, 2009*). | *Synonyms:* economic resources (*Simberloff, 2013*); availability of funds, available budget (*Dana et al., 2019*), resources available (*Sohrabi et al., 2020*) *Similar terms:* "availability of funds is guaranteed during the necessary time frame to achieve the specific IAS management objective" (*Dana et al., 2019*); "Is funding for core operations secure for at least 2 years, and the project has undertaken the necessary financial planning and achieved partial success in developing sources of long-term funding to sustain core costs for the next 5 years?" (*Corbin et al., 2017*) |
| | 23. Economic and social relevance of target species | Economic significance of the species and social reception of eradication action. This includes: (a) the possibility to abandon various benefits of using the species, (b) social pressure to stop or intensify eradication due to cultural or health reasons. | *Similar terms:* social context (*Crowley, Hinchlife & McDonald, 2017*), social perception (*Dana et al., 2019*), level of public awareness (*Spring & Cacho, 2015*), "is it highly likely that social or political resistance to control will lead to project failure?" (*Corbin et al., 2017*), sociopolitical factors (*Dodd et al., 2015*) *Related term:* nil cultivation value (*Dodd et al., 2015*) |
| **Reinvasion context** | 24. Invasion pathways | Possibilities of preventing the reappearance of an invasive species after eradication. This includes: (a) analysis of vectors and pathways enabling species reinvasion (*e.g.,* propagule pressure studies), (b) possibilities of blocking the potential pathways of invasion by *e.g.,* phytosanitary regulations. | *Synonym:* risk of reinvasion (*Dana et al., 2019*), likelihood of reinvasion (*Booy et al., 2017*) *Similar terms:* prevention of reinvasion (*Dodd et al., 2015*), sanitary control (*Pluess et al., 2012a*) *Related term:* probability of reinvasion (*Simberloff, 2003*) |

included '**isolation of infestation**' (Table 1, factor 3), which describes geographical and ecological isolation of the target population. This factor was mentioned earlier as 'insularity' by *Pluess et al. (2012a)* and *Pluess et al. (2012b)*.

Factors mentioned earlier in the literature but not incorporated in USEF are included in Table 1 only as similar or related terms for comparison purposes. Similar terms were differently defined in their original publications than in our system, but conveyed a similar meaning. Factors, which in our opinion had a lower significance for eradication success or their impact is difficult to assess, were evaluated as related terms. For example for the factor '**number and distribution of propagules**' (*i.e.,* size and spatial structure of propagule

**Table 2  Scoring of eradication related factors included in Unified System for assessing Eradication Feasibility (USEF).**

| Group | Factor | Score value | Description of categories for scoring | Score concept credit |
|---|---|---|---|---|
| **Size and location** | 1. Infestation size | 1 | Less than 1 ha, | Scale used by *Blood & James* (*2018*, slightly changed), *Blood et al.* (*2019*, slightly changed) to describe monitoring area size |
| | | 2 | 1 to 10 ha, | |
| | | 3 | Between 10 and 100 ha, | |
| | | 4 | Between 100 and 1,000 ha, | |
| | | 5 | More than 1,000 ha. | |
| | 2. Number of separate infestation sites | 1 | One infestation site, | Extended scale used by *e.g., Cunningham et al. (2004), Panetta & Timmins (2004), Blood & James (2018)* and *Blood et al. (2019)* |
| | | 2 | 2 or 3 infestation sites, | |
| | | 3 | 4 or 5 infestation sites, | |
| | | 4 | Between 6 and 10 infestation sites, | |
| | | 5 | More than 10 infestation sites. | |
| | 3. Isolation of infestation | 1 | Island (area less than 2,000 $km^2$) located more than 500 km from the nearest continent, | Original concept and definitions |
| | | 2 | Island (area less than 2,000 $km^2$) located 500 km to 10 km from the nearest continent, | |
| | | 3 | Island (area less than 2,000 $km^2$) located less than 10 km from the nearest continent, | |
| | | 4 | Mainland (continent or island more than 2,000 $km^2$) surrounded by distinct ecological barriers, | |
| | | 5 | Mainland (continent or island more than 2,000 $km^2$), no distinct ecological barriers. | |
| | 4. Monitoring area size | 1 | Less than 1 ha, | *Blood & James* (*2018*, slightly changed), *Blood et al.* (*2019*, slightly changed) |
| | | 2 | Between 1 and 10 ha, | |
| | | 3 | Between 10 and 100 ha, | |
| | | 4 | Between 100 and 1,000 ha, | |
| | | 5 | More than 1,000 ha. | |
| | 5. Monitoring rate | | Number of required visits during the year: | Original concept |
| | | 1 | One visit, | |
| | | 2 | 2 or 3 visits, | |
| | | 3 | 4 or 5 visits, | |

**Table 2** (*continued*)

| Group | Factor | Score value | Description of categories for scoring | Score concept credit |
|---|---|---|---|---|
| | | 4 | Between 6 and 12 visits, | |
| | | 5 | More than 12 visits. | |
| **Ease of access** | 6. Land use and ownership of infested area | | Accessibility resulting from land ownership relations: | Partly after *Pluess et al. (2012a)* and *Dana et al. (2019)* |
| | | 1 | Low complexity of the management mosaic, no difficulties in accessing the target area; | |
| | | 2 | High complexity of the management mosaic, obtaining permits from private landowners and/or administrative requirements takes effort; | |
| | | 3 | High complexity of the management mosaic, administrative requirements not available and/or access to private properties is problematic. | |
| | 7. Accessibility | | Accessibility resulting from landform and distance to infestation: | Partly after *Blood & James (2018)* and *Blood et al. (2019)* |
| | | 1 | The distance to be traveled by the workers is small and the topography favors the action, | |
| | | 2 | Difficulties in accessing the target area, once workers are there all infestation sites easily accessible, | |
| | | 3 | Distance to be traveled moderate and most infestation sites readily accessible, | |
| | | 4 | Distance to be traveled moderate and/or most infestation sites difficult to access, | |
| | | 5 | Distance to be traveled great and all infestation sites difficult to access. | |
| **Fitness and fecundity** | 8. Adaptation to new climate conditions | | Factor composed of species climate adaptability (low—species present in 1 climatic zone, medium—present in 2 zones, high—present in 3 or more climatic zones) and climate distance between continuous (native and/or non-native) species range and infestation location (short—species present in the same climatic zone, medium—species present one climatic zone away, high—species present two or more climatic zones away). | Original concept |
| | | 1 | Low adaptability and high distance, | |
| | | 2 | Low adaptability and medium distance or medium adaptability and high distance, | |
| | | 3 | High adaptability and high distance or medium adaptability and medium distance, | |
| | | 4 | High adaptability and medium distance, | |
| | | 5 | Short climate distance regardless of species adaptability. | |

**Table 2** (*continued*)

| Group | Factor | Score value | Description of categories for scoring | Score concept credit |
|---|---|---|---|---|
| | 9. Number and distribution of propagules | 1 | Less than 1,000 propagules per m² with concentrated distribution, | Original concept |
| | | 2 | Between 1,000 and 10,000 propagules per m² with concentrated distribution, | |
| | | 3 | Less than 1,000 propagules per m² with dispersed distribution, | |
| | | 4 | Between 1,000 and 10,000 propagules per m² with dispersed distribution, | |
| | | 5 | More than 10,000 propagules per m² regardless of distribution. | |
| | 10. Vegetative propagation | | Production of vegetative propagules: | Original definitions |
| | | 1 | The plant does not reproduce vegetatively; | |
| | | 2 | Vegetative propagation very rare, only based on regenerative capabilities; | |
| | | 3 | Production of vegetative propagules not common; | |
| | | 4 | Production of vegetative propagules common. | |
| | 11. Propagule longevity | | Maximum longevity of propagules: | Definitions after *Thompson, Bakker & Bekker (1997)* |
| | | 1 | Less than 1 year, | |
| | | 2 | Between 1 and 5 years, | |
| | | 3 | More than 5 years. | |
| | 12. Pre-reproductive period | | Minimum pre-reproductive period: | Extended scale used by *Panetta & Timmins (2004)*, *Panetta (2016)*, *Blood & James (2018)*, *Blood, James & Panetta (2018)* and *Blood et al. (2019)* |
| | | 1 | More than 10 years, | |
| | | 2 | Between 2 and 10 years, | |
| | | 3 | Between 1 and 2 years, | |
| | | 4 | Between 1 and 12 months, | |
| | | 5 | Less than 1 month. | |
| **Detectability** | 13. Detection possibility | | The target species: | Partly after *Panetta & Timmins (2004)* |
| | | 1 | Emergent and with distinctive features, identifiable from a distance greater than 1,000 m, remote identification possible; | |
| | | 2 | Emergent and with distinctive features, identifiable from a distance 2–1,000 m; | |
| | | 3 | Either emergent or with distinctive features, identifiable from a distance <2 m; | |
| | | 4 | Non-emergent from vegetation and with no distinctive features (*e.g.,* for identification a magnifier is needed); | |
| | | 5 | Impossible to distinct from other organisms without special equipment. | |

**Table 2** (*continued*)

| Group | Factor | Score value | Description of categories for scoring | Score concept credit |
|---|---|---|---|---|
| | 14. Annual period of detectability prior to seed set | 1 | More than 9 months, | Extended scale used by *Panetta & Timmins (2004)*, *Blood & James (2018)* and *Blood et al. (2019)* to describe detectability period |
| | | 2 | Between 6 and 9 months, | |
| | | 3 | Between 3 and 6 months, | |
| | | 4 | Between 1 and 3 months, | |
| | | 5 | Less than 1 month. | |
| **Knowledge** | 15. Knowledge of current location of infestation sites | 1 | Area under consideration for eradication well investigated, location of all of infestation sites well known, distribution maps have been made; | Original definitions |
| | | 2 | Location of infestation sites not well known and/or distribution maps not available; | |
| | | 3 | Location of only the largest sites known. | |
| | 16. Understanding of species biology | 1 | Local population characteristics well known, | Original definitions |
| | | 2 | Only general knowledge of the biology of the species available from other environmental conditions | |
| | | 3 | Poor knowledge about the biology of the species. | |
| | 17. Eradication achieved elsewhere | 1 | Successful actions carried out in different environmental conditions, more favorable to invasion; | Original definitions |
| | | 2 | Successful actions carried out in similar conditions, but there was no successful action in conditions less favorable to invasion; | |
| | | 3 | Successful actions carried out in different conditions, less favorable to invasion; | |
| | | 4 | So far there have been no attempts to eradicate target species or all other eradication attempts have been unsuccessful. | |
| **Cognition and resources** | 18. Reaction time | | Time elapsing from species detection to eradication start: | Extended scale by *Panetta (2016)* and *Blood & James (2018)*, used to describe residence time |
| | | 1 | Less than 1 year, | |
| | | 2 | 1 to 2 years, | |
| | | 3 | Between 2 and 5 years, | |
| | | 4 | Between 5 and 10 years, | |
| | | 5 | More than 10 years. | |

**Table 2** (*continued*)

| Group | Factor | Score value | Description of categories for scoring | Score concept credit |
|---|---|---|---|---|
| | 19. Applicable control methods | | *Step 1.* Assess the effectiveness of individual methods (physical control, cultural control, chemical control, biological control, sanitary control, other control methods) using a 4 step scale: very effective, moderately effective, ineffective, not applicable or has not been tested; *Step 2.* Score the impact of the entire factor: | Original concept |
| | | 1 | At least one method proved to be very effective; | |
| | | 2 | No method proved to be very effective, but at least one method proved to be moderately effective; | |
| | | 3 | No effective methods known. | |
| | 20. Personnel awareness | 1 | Awareness and enthusiasm of all workers sufficient, | Original definitions |
| | | 2 | Awareness person-dependent and/or varies over time, | |
| | | 3 | Some workers show discouragement and/or a lack of understanding of the need to combat the invasion. | |
| | 21. Coordination between monitoring agencies | 1 | One plenipotent institution involved in the action, | Original definitions |
| | | 2 | All involved parties coordinate easily, | |
| | | 3 | Involved institutions generally cooperate but sometimes problems are arising that stop the communication flow, | |
| | | 4 | Several institutions involved in the action that do not cooperate with each other. | |
| | 22. Sufficient allocation of resources | 1 | Resources guaranteed at a level appropriate to the needs, available for the whole duration of the project; | Definitions partly after *Simberloff (2013)* and *Dana et al. (2019)* |
| | | 2 | Financing less than sufficient, but stable or action possible due to voluntary workers engagement; | |
| | | 3 | Guaranteed for at least 2–5 years and it is possible to supplement them; | |
| | | 4 | Inadequate budget, insufficient duration of the financing, possibility of their supplementation unknown. | |

| Group | Factor | Score value | Description of categories for scoring | Score concept credit |
|---|---|---|---|---|
| | 23. Economic and social relevance of target species | | Expected public attitude to the eradication action: | Concept partly after from *Dana et al. (2019)* |
| | | 1 | Public support and high awareness of the negative effects of the invasion, | |
| | | 2 | No public support or opposition, | |
| | | 3 | Public opposition, due to *e.g.,* economic, cultural or health reasons. | |
| **Reinvasion context** | 24. Invasion pathways | 1 | Potential pathways of invasion limited or at least partially blocked by phytosanitary regulations, | Original concept and definitions |
| | | 2 | Various pathways of invasion and difficult to manage, | |
| | | 3 | Potential pathways of invasion have not been identified. | |

store, see definition of factor 9 in Table 1) the related terms include 'propagule dispersal' (*Blood et al., 2019*) and 'mode of propagule dispersal' (*Panetta & Timmins, 2004*). With no doubt propagule dispersal has a major impact on the invasion success. Seed dispersal is usually mediated through several dispersal mechanisms involving primary and secondary dispersal and natural or human assisted dispersal. Indicating a direct relation between predominant propagule dispersal mechanism of a given species and success of eradication of target population may be difficult to evaluate because of that. *Blood et al. (2019)* stated that ''human-mediated dispersal should be easier to influence''. However evaluating the ''potential for managing propagule dispersal'' (*Blood et al., 2019*) in practice can be very difficult. The definition of an invasive species states that its appearance in a new site is mediated through human actions (see definition of invasive species by *e.g.*, *Blood et al., 2019*). Another difficulty is demonstrating which of the dispersal mechanisms exploited by an invasive species was responsible for another site establishment. Such establishment instances are usually rare events, which are very difficult to observe and may be driven by many different dispersal mechanisms. Therefore we selected propagule distribution, which is a measurable consequence of dispersal and indicates individual recruitment location. In our system we did not include all the traits associated with species biology that may affect the success of the invasion. As separate factors we included only those traits that directly affect the success of the invasion in a significant way (see factors from fitness and fecundity group, Table 1). It is worth emphasizing here that the requirement of detailed study of the biology of the invasive population has been included in factor 5, 'understanding of species biology' (Table 1).

## Description and interpretation of factors selected for the system

The synthetic characteristic of all the factors included in USEF is presented in Table 1. For some factors additional explanations/comments were necessary. In the text below we have analyzed in more detail only the most ambiguous and controversial factors leaving the

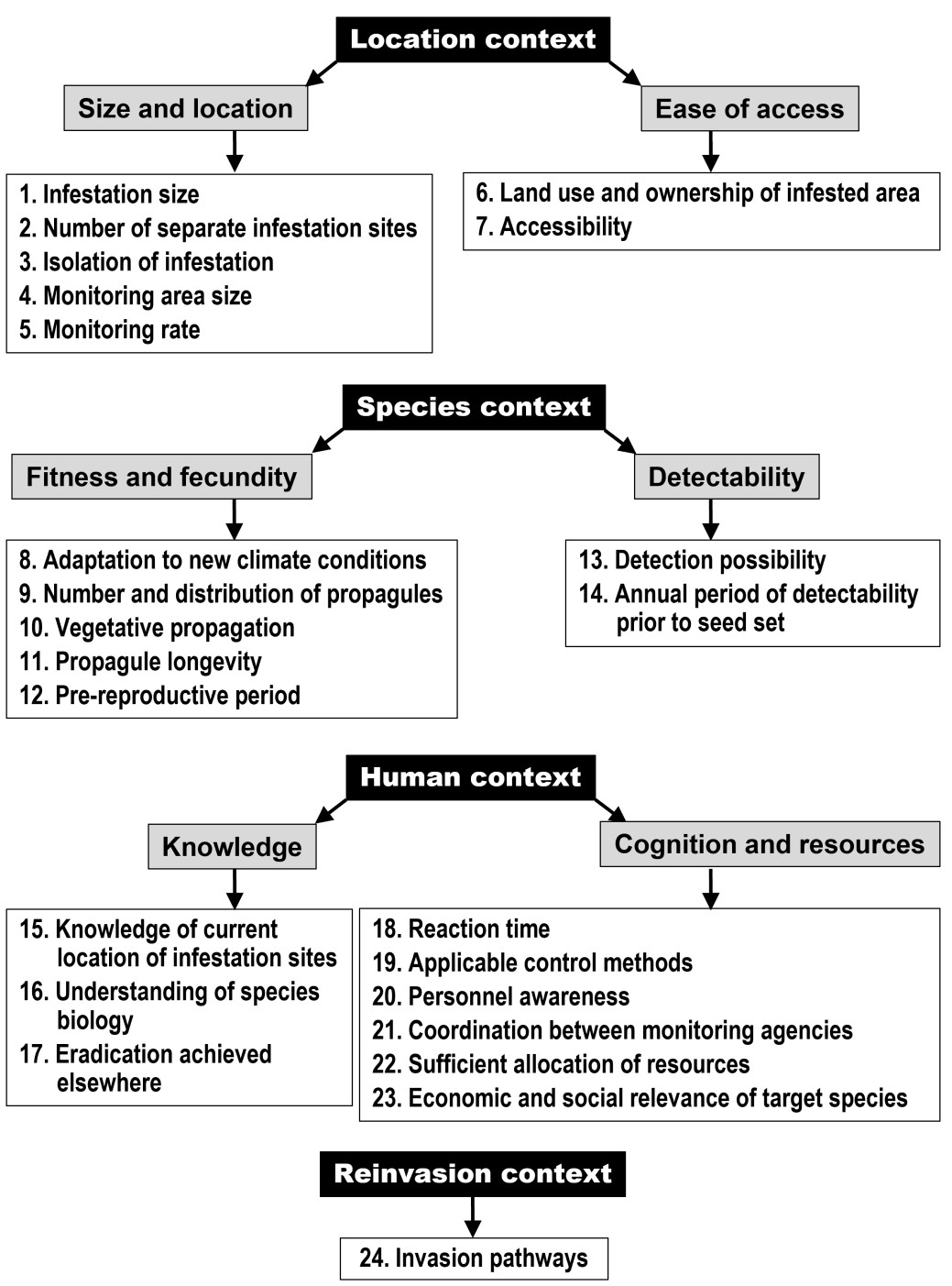

**Figure 1** Structure diagram of the Unified System for assessing Eradication Feasibility (USEF) showing the classification of factors affecting the probability of eradication of an alien plant species.

more obvious ones without a comment. The text is organized in accordance with Table 1 and Fig. 1 and the subsections correspond to specific factor groups.

## Size and location of target population

One of the most important factors influencing eradication feasibility is the size of the target population. The most universal way of presenting this factor is reporting the area ($m^2$, ha, $km^2$, etc.) occupied by the invasive population, therefore we propose to name this factor '**infestation size**' (see factor 1 in Table 1). Distinguishing 'net infestation size' and 'gross infestation size' (*Rejmánek & Pitcairn, 2002*; *Dodd et al., 2015*; *Blood et al., 2019*) enables including the spatial organization of the population, but if the population is not concentrated (individuals are sparsely scattered within the infested area) the term 'gross infestation size' may be misleading. We therefore propose to rename the 'gross infestation size' to '**monitoring area size**' (factor 4, Table 1) as it does not necessarily represent the infested area, but an area which needs to be monitored for the species (re)appearance.

The aim of eradication is to diminish the population size lowering the '**infestation size**' and '**number of separate infestation sites**' until they reach ''0'' (Table 1, factors 1 and 2, respectively). Ideally the values of these factors should not remain constant during the eradication process. We thus propose to include information on the stage of eradication, *e.g.*, time elapsing from the start of eradication (*e.g.*, initial infestation size, number of separate infestation sites after 5 years of eradication campaign). We believe that if infestation size is accompanied by information on specific eradication stage (*i.e.,* initial pilot actions, beginning of eradication, 5 years after eradication commenced, etc.) the analysis of such eradication process and its comparison with actions performed elsewhere will be much easier than without this information clearly presented. The addition of time dimension also allows reporting and comparing 'infestation size' in different time points during the eradication process. In contrast to the variable 'infestation size' the size of the monitored area should not change (with exception of an unsuccessful eradication only increasing the size). The whole area defined by '**monitoring area size**' (Table 1, factor 4) needs to be constantly monitored for reappearance of the invasive species, as stated by *Burgman et al. (2013)* ''total infested area can never decrease''. It is also advisable to add a ''buffer zone'' to the monitoring area, to contain the spread of the species (*Fletcher et al., 2015*).

Classifying '**monitoring rate**' into one of the groups was not straightforward (factor 5, Fig. 1). We decided to include this factor in the location context and size and location group, because both components of this factor (annual monitoring period and recommended monitoring intensity) are mostly determined by environmental characteristics of the non-native range (Fig. 1 and Table 1), which may be perceived superior to species context factors.

## Ease of access to target population

It is beyond all doubt that ''Land use directly affects the invasion process because it modifies disturbance regimes and environmental conditions'' (*Pauchard & Alaback, 2004*; *Epanchin-Niell et al., 2010*; *Epanchin-Niell et al., 2012*). However 'land use' as an eradication feasibility

factor is difficult to measure, differently understood and used in different ways to predict the probability of eradication success. *Dodd et al. (2015)* distinguished two states of 'land use', *i.e.,* residential land use and environmental land use, assigning respectively high and low expected relationship with eradication feasibility. *Panetta (2015)* mentioned that different eradication costs are associated with different land use. *Panetta & Timmins (2004)* mentioned land use in the accessibility to infestation context. Nevertheless neither of these authors attempted to define 'land use' in eradication feasibility context. In contrast, *Pluess et al. (2012a)* did not use the term 'land use', but distinguished three categories of accessibility to the infestation depending on the level of difficulties considering accessing the target area: "access to private properties problematic, remote areas concerned"; "some difficulties to access the target area"; "no difficulties to access the target area". These authors also distinguished a factor connected with land use—'man-made habitat' including it in the 'location specific factors' group (*Pluess et al., 2012a*). *Rohal, Cranney & Kettenring (2019)* wrote about 'landscape factors' associated with human disturbance.

Additional problems and misunderstandings associated with the term 'land use' may have originated from the fact that *Dodd et al. (2015)* used the term 'general accessibility' to describe "distance to manager" and proposed another factor 'number of separate land managers' to describe the ownership relations of the infested area. It is worth mentioning, that this last factor is directly linked with possible difficulties with the access to the infestation resulting from ownership relations of infested area. *Epanchin-Niell et al. (2010)* stated that land subdivision into many ownerships impairs invasion control. They also introduced the term 'management mosaics' ("landscapes comprising many individually managed properties with a variety of uses", *Epanchin-Niell et al., 2010*). Considering all the different perceptions of land use mentioned above we propose a longer, but more precise factor name—'**land use and ownership of infested area**'. We further propose to subdivide this factor into four components describing different ways of its influence on the eradication process (factor 6, Table 1).

## Fitness and fecundity of the target population

The adaptation of the invasive organism to the climatic conditions in a new environment (*e.g.,* *Galera, Chwedorzewska & Wódkiewicz, 2015*) is as important as land use. *Dodd et al. (2015)* proposed a factor named 'climate match of species with site'. This factor could be used to prioritize eradication efforts of especially dangerous species (candidate species), which have not reached the status of invasive species in a specific area (*Cunningham et al., 2004*; *Branquart et al., 2016*) however its use for the evaluation of eradication feasibility is in our opinion doubtful. If the target species has already established a naturalized and/or invasive population in an area the analysis of this factor is in our opinion not necessary. The ability to exploit specific climatic conditions may not be uniform across the whole species and may be population specific, due to intra- and inter-specific admixture (*Lachmuth, Durka & Schurr, 2010*; *Lawson Handley et al., 2011*). The species has to be highly climate adapted to be considered naturalized and even more to be declared invasive (*e.g.,* *Richardson et al., 2000*; *Richardson, Pyšek & Carlton, 2011*; *Colautti & MacIsaac, 2004*). Additionally, climate match is difficult to assess as many different climatic factors may interplay with

different species life history traits (*Kriticos & Brunel, 2016*; *Galera et al., 2018*). The use of several map types illustrating the distribution of climate types in the species native and non-native range may be an acceptable solution (*Branquart et al., 2016*). Nevertheless such traits of invasive species as phenotypic plasticity (*Galera, Chwedorzewska & Wódkiewicz, 2015*), tolerance to a broad range of environmental conditions (*Higgins & Richardson, 2014*) and rapid adaptation to climate conditions (*Colautti & Barrett, 2013*), that all blur the climatic match of the species, have to be considered as well. Therefore we decided to name this factor '**adaptation to new climate conditions**' and to distinguish its two components (see factor 8, Table 1).

Another factor from the group aggregating species traits is 'vegetative propagation' (factor 10, Table 1). *Dodd et al. (2015)* and *Blood et al. (2019)* named this factor 'vegetative reproduction', but we propose to change its name to '**vegetative propagation**' as no new genetic combination arises through asexually driven population number increase. As a result there is no chance of new combination of alleles to arise (save mutations) with a new invasion potential. If the alien species invades a habitat differing much from the home habitat, change in genetic information due to new allele combination through intra- or inter-specific admixture usually triggers a rapid spread of the invasive species in the non-native range (*e.g.*, *Buhk & Thielsch, 2015*).

## Detectability of target species

The factor describing the detectability of the focus species, '**detection possibility**' (Table 1, factor 13), also needs to be clarified as it has different aspects addressed in the literature. This factor describes the feasibility of correctly identifying the focus species. The important aspect here is the ability to distinguish the invader from other, local species to avoid mistakes. Many means of such detection are possible, depending on the species. Besides identification with naked eye or microscope (*Pluess et al., 2012a*), detection using remote sensing (*Müllerová et al., 2016* and literature cited therein) or identification by detector dogs (*Cherry et al., 2016*; *Williams et al., 2019*) may be considered.

## Reinvasion possibility

In a separate group, the "reinvasion context", we have included one factor, the '**invasion pathways**' (factor 24, Fig. 1) as repeated invasion brings the necessity to repeat the eradication action. For this factor it is important to consider and analyze the invasion pathways and vectors which may reintroduce the unwanted species after successful eradication or reinforce the population being eradicated (*e.g.*, *Faulkner et al., 2020*; *Shackleton et al., 2020*).

The inclusion of other factors influencing eradication success into USEF is possible. However in every instance we suggest giving a precise definition and discussing the differences from previously used terms.

## Evaluating eradication feasibility

Different values/states of each factor may either promote or restrict eradication. We analyzed each factor separately and within it distinguished from 3 to 5 categories (depending on factor) potentially influencing the eradication success in a different way. We assumed
that the distinguished categories should be universal (applicable in the assessment of each invasive species management action) and relatively easy to classify (categorization should be clear and mutually exclusive). We used an ordinal scale to score the factor's category impact on eradication success (Table 2). For all factors the lowest score values (1) reflect factor states promoting eradication and highest score values (maximum for each factor) reflect factor states hindering eradication most.

In order to define and score factor states we tried to use the suggestions given in the literature, possibly clarifying and adjusting the sets of categories to all possible cases (Table 2). Nevertheless, only for three factors ('infestation size', 'monitoring area size', 'propagule longevity') we decided to adopt a scale proposed by other authors. For three other factors ('number of separate infestation sites', 'pre-reproductive period', 'reaction time') we extended the scales and proposed new factor value definitions. We proposed completely new definitions for the categories distinguished for the following twelve factors: 'isolation of infestation', 'monitoring rate', 'adaptation to new climate conditions', 'number and distribution of propagules', 'vegetative propagation', 'detection possibility', 'knowledge of current location of infestation sites', 'understanding of species biology', 'eradication achieved elsewhere', 'applicable control methods', 'personnel awareness' and 'invasion pathways'. In all other cases, our concepts and/or definitions of categories are partly based on the literature (Table 2).

Some factors were readily dividable into ranges, which could be easily ranked. Other factors were more problematic, raising the necessity to develop a more complicated approach. Below we focus on the more demanding factors in the sense of categorization and/or scoring.

For '**isolation of infestation**' (Table 2, categories for factor 3) we used a very simple categorization proposed by *Pluess et al. (2012a)* and *Pluess et al. (2012b)*. In addition we took into account the importance of geographic (distance of islands from the nearest continent) and ecological barriers within the mainland (*i.e.,* continent or large island). We compiled the ranges of the islands' size and distances based on the variability of islands in terms of these characteristics (according to data in UN Islands Directory, *Dahl, 1998*). In the case of islands located in the Southern Ocean, we propose to take into account their distance from a continent other than Antarctica (this continent is a marginal source of plant propagules, *Galera et al., 2018*).

In the case of '**monitoring rate**' we assumed that a typical search attendance is once a month. If the recommended monitoring frequency is higher than once a month, and the monitoring season lasts the entire year, then a very high monitoring intensity with over 12 visits per year is needed (Table 2, factor 5).

For '**land use and ownership of infested area**' we found it impossible to create clear categories with the use of all four components (compare definition of factor 6. in Table 1 and definitions of categories of this factor in Table 2). We decided to include land ownership relations, because without obtaining the appropriate approvals, the action would be illegal. The second argument is the fact that the data on the number and attitude of owners of the area under consideration for species eradication is relatively easy to obtain. For the factor of site '**accessibility**' (factor 7, Table 2), we have tried to include both the distance from
the headquarters that workers have to travel to the infestation site and possible difficulties resulting from landform.

The factor '**adaptation to new climate conditions**' (factor 8, Table 2) is an interplay between the climatic conditions in the native and nonnative range and species adaptability to new climate conditions. We decided to measure both species adaptability and climate distance based on five major climate zones distinguished in a popular and constantly updated Köppen-Geiger classification (*e.g.*, *Peel, Finlayson & McMahon, 2007*; *Beck et al., 2018*; *MacLeod & Korycińska, 2019*). For simplicity we categorized species climate adaptability into: low—species present in one climatic zone, medium—present in two zones, high—present in three or more climatic zones. In the case of climate adaptability, both primary and secondary geographical range of the species should be taken into account. For new alien taxa, *e.g.*, hybrids/cultivars having a relatively small range, it should be assumed that their climatic adaptation abilities are unknown. We categorized the climatic distance between the site of the analyzed infestation and the most similar climatic conditions within the continuous range of the species as: short—species present in the same climatic zone, medium—species present one climatic zone away, high—species present two or more climatic zones away. Subsequently we scored the possible combinations of these variables. Our rationale for scoring was that the more the species is adaptable and the shorter the climatic distance the more difficult it will be to successfully eradicate the species.

In '**number and distribution of propagules**' (categories for factor 9 in Table 2), both seeds and vegetative propagules should be taken into account. We assume that if propagule number is lower and they are concentrated in specific locations it is easier to deplete the propagule bank. The cutoff values for propagule number were based on the assessment of the size of soil seed banks of different plant communities presented in *Baskin & Baskin (2001)*, as seeds may retain viability much longer than vegetative propagules.

For '**propagule longevity**' (factor 11, Table 2) we decided on the simple 3-point scale, taking into account the concept of three types of seed banks (*Thompson, Bakker & Bekker, 1997*). We are also aware that it is very difficult to determine the seed bank longevity greater than 5 years. In order to determine this factor, longevity of both generative and vegetative propagules should be considered. When classifying into a category, the longest period should be taken into account. In practice it is usually seed longevity.

Similarly to factor 11, both generative and vegetative propagules must be considered in determining the '**pre-reproductive period**'. When classifying into categories, the shortest period should be taken into account (in practice, vegetative reproduction usually occurs faster). The scale of 'pre-reproductive period' (factor 12, Table 2) was extended by us to include species, that start producing propagules after more than 10 years (*e.g.*, some trees).

The four-point scale used by *Panetta & Timmins (2004)*, *Blood & James (2018)* and *Blood et al. (2019)* for the detectability period has been adopted by us to categorize '**annual period of detectability prior to seed set**' and extended to six levels. We classified species which are "always inconspicuous" as plants with annual period of detectability of "less than 1 month" (factor 14, Table 2). Only sexual reproduction has to be considered when categorizing, as vegetative propagules are often produced all year round.
To assess the impact of '**eradication achieved elsewhere**' on the target eradication it is important to know if any of them proved to be successful. Gains from successful eradications depend on the climatic difference between the location of those actions and the location of eradication attempt being analyzed. This is reflected in our evaluation of the factor categories on eradication success (Table 2).

The evaluation of '**applicable control methods**' includes a two-step procedure. The first step should be to assess if any of the known eradication methods is successful. The second step is scoring the factor based on the most successful method. The rationale is that at least one effective eradication method suffices to remove the species.

In the case of '**knowledge of current location of infestation sites**' and '**understanding of species biology**', we assume that the managers have a basic knowledge of the location of the target infestation and biology of the species (compare Table 2, categories for factors 15 and 16). When assessing the level of '**personnel awareness**' and '**sufficient allocation of resources**' (Table 2, factors 20 and 22), only three categories were proposed, taking into account the fact that the precise definition of categories is not possible. We are also aware that objective self-assessment is difficult, if not impossible sometimes. We assume the good will and common sense of workers rejecting the possibility of a total lack of awareness and financing. In the case of '**coordination between monitoring agencies**' (Table 2, categorization for factor 21), we assumed that the situation in which the action is carried out by one institution is the most favorable (*Dodd et al., 2015*).

For '**invasion pathways**', we assumed that the risk of reinvasion always exists. When categorizing, we took into account the knowledge of potential introduction pathways and the possibility of blocking or controlling them (compare Table 2, categories for factor 24).

## Indications for the system usage

It is tempting to sum up all the score values to depict the overall effectiveness of an eradication action. Unfortunately because of the different nature and varying impact of factors on eradication success this cannot be done. Even if all the factors are scored, the resulting comparison between actions may be misleading. To evaluate the overall feasibility of an analyzed eradication action the factors should be weighted according to their importance in obstructing the eradication success. This can be achieved only after sufficient data accumulation. This data should be reported according to a unified reporting scheme. Our work is a suggestion of such a scheme.

At first the quality of the assessment should be performed. The higher the number of factors, which cannot be evaluated due to the lack of information the lower the quality of the assessment. The lack of accumulated data acquired in accordance with a unified scheme is the main obstacle to comparing eradication actions (*e.g.*, *Wilson et al., 2014*). *Cunningham et al. (2004)* noted: "The eradication feasibility measure reported in this study needs further testing against actual eradications". Despite the passage of almost twenty years since their publication, this remark is still relevant. The number of cases of eradication actions, the effectiveness of which has already been proven, remains small. For example *Gardener, Atkinson & Rentería (2010)* out of 30 analyzed eradication projects involving

23 species evaluated only four species as eradicated. *Dodd et al. (2015)* out of 17 analyzed infestations deemed only one species as successfully eradicated.

At present to circumvent the necessity of weighting factors we propose first to divide the scored value by the maximum value for each factor. In such a way the evaluation of each factor's input is represented by a number between 0.20 and 1.00. The sum of factors' inputs results in an overall eradication action feasibility assessment. The greater the number the more difficult it is to eradicate the species.

Another problem associated with eradication evaluation is missing information. This is understandable, especially in the case of recent actions as it is sometimes difficult at first to ascribe a value to an eradication related factor. These gaps however impair our ability to evaluate the action. An earlier proposed method is to ascribe a mean value (*Weiss & Iaconis, 2002*) or a default value in absence of information (*Panetta & Timmins, 2004*; *Blood & James, 2018*; *Blood et al., 2019*). This attempt however yields data for comparison, but the quality of such an assessment or comparison is low. Therefore we propose to restrain from comparison until some assessment of data is available. The drawbacks associated with lack of data are further discussed in the next section.

## An example of USEF usage

To illustrate how our system works, we prepared an exemplary comparison of three different invasions (Table S1 in Supplemental Data). It is a thorough comparison of a *P. annua* invasion in the Antarctic on King George Island, and *Stellaria media* (L.) Vill. and *P. annua* invasions on Macquarie Island. We chose these invasions to show how our system can help to compare invasions of two different species at the same site, or invasions of the same species but in different locations. We assessed each factor using our proposed scale and added descriptions and explanations for each assessment based on available data published in the literature.

We summed the factors' inputs for all three eradications. The *P. annua* eradication on King George Island was evaluated at 13.75, *P. annua* on Macquarie Island at 8.38 and *S. media*—at 12.86. While we were able to evaluate all the 24 factors for *P. annua* on King George Island, we found no information for evaluating 10 factors for *P. annua* on Macquarie Island and five factors for *S. media* on this island. This stresses the importance of missing data. If we scored each of the factors with missing data as 0.50, the results for *P. annua* and *S. media* on Macquarie Island would be respectively 13.38 and 15.36, but at this level of missing data the eradication assessment has a low quality and the whole eradication action comparison needs to be treated with caution as stated earlier. We can further compare only separate and known factors between the selected eradication actions.

One could assume that in every eradication instance managers possessed information on the infestation size, but the lack of reporting of this crucial information disables further sound comparisons. Even in the example of relatively well documented invasions on Macquarie Island we were forced to mark some factors as unknown, because specific information was insufficient with just vague descriptions published or even lacking completely (Table S1, factors 1, 2, 4). The absence of almost any information concerning infestation size and distribution of *P. annua* on Macquarie Island severely affected the

eradication success probability assessment (see also factor 15 in Table S1). The only published descriptions are very general and no studies about the topic were published to inform about the exact area.

This shows how remote assessment is difficult and highlights how much data is still needed. For the comparison of different eradication campaigns, comparable datasets organized in accordance to commonly defined driving factors are needed. Unfortunately in up-to-date literature even the most evident and necessary data is not presented. For example *Pluess et al. (2012b)* observed that "precise quantitative measurement of the infestation area (in $km^2$) was rarely available".

For the invasions compared here as well as for other islands, the distance from the closest continent/landmass (*e.g.*, *Tye, Soria & Gardener, 2002*; *Caujape-Castells et al., 2010*; *Kueffer et al., 2010*) seems to be a good measure of the degree of isolation. However, even in the case of such an easily measurable factor, the assessment must take into account the specificity of the location of a given infestation. Macquarie Island lies 1,800 km form Antarctica (nearest continent, *Dahl, 1998*), 1,100 km from New Zealand and 640 km from Auckland Islands and is surrounded by a few small islets (*Macquarie Island Nature Reserve and World Heritage Area, 2006*). King George Island belongs to the South Shetlands archipelago and is situated only 160 km from Antarctica. For this reason, the isolation index (based on distances to nearest island, group and continent) proposed by Dahl, is high for Macquarie Island (112), while for King George Island it is only 26 and the island was not considered "more isolated" (*Dahl, 1998*). However, Antarctica is a marginal source of propagules, and the ecological isolation of the Antarctic region is enhanced by the atmospheric Polar Vortex and oceanic Polar Frontal Zone (*Galera et al., 2018*). In line with our earlier proposal (see chapter Evaluating eradication feasibility), we took into account the distance of two analyzed islands from another continents other than Antarctica. Taking into account that the terrestrial ecosystems of King George Island are environmental islands (Table S1), it can be concluded that this invasive population of *P. annua* is one of the world's most isolated infestations. In this situation, we considered the isolation of both analyzed islands to be a very favorable factor for eradication and scored them in the same way (Table S1, factor 3).

Our scoring of the accessibility factor differed between the compared eradications. The topography of Macquarie Island makes it difficult to reach the invasive populations, and access may be also restricted by dense seal colonies and penguin rookeries. Such problems are non-existent on King George Island, as the target populations are easily reached from the buildings of the Polish Antarctic Station (Table S1, factor 7).

Unfortunately we were not able to escape all ambiguity by using our system, even if precise information had been published. This is depicted *e.g.*, in our assessment of the number of separate infestation sites in the case of *S. media* on Macquarie Island. *Williams et al. (2019)* wrote about "eight loosely defined, high density populations which varied in area and *S. media* density" and "six isolated plants [...] detected outside these populations". We considered this infestation to consist of eight sites, disregarding information on separate individuals (Table S1, factor 2).

In comparison to the two infestations on Macquarie Island, the *P. annua* invasion on King George Island has either a greater or equal chance of being successfully eliminated.

Even with the simplified scoring system we used, the analysis of Table S1 allows specific conclusions to be drawn. The compared cases are similar for eight factors as the same factors' inputs were obtained for all three infestations (factors 3, 6, 8, 13, 16, 18, 23 and 24 in Table S1). This is understandable because the two islands are located in the polar climate zone, and both *P. annua* and *S. media* are small annual weeds with a similar life strategy.

The comparison of two *P. annua* populations according to ''one species/different regions'' scheme shows that there is a greater possibility of successful eradication of the species from King George Island in terms of three factors: 'number and distribution of propagules', 'knowledge of current location of infestation sites' and 'eradication achieved elsewhere' (factors 9, 15 and 17 in Table S1, respectively). On the other hand, 'propagule longevity' (factor 11) reduces the likelihood of successful eradication of *P. annua* from this Island. It is worth noting that in terms of 'propagule longevity' and 'number and distribution of propagules', the two compared populations of annual bluegrass differ significantly. The invasive *P. annua* population on King George Island can theoretically be considered relatively easy to eradicate. However, in practice, even this so far turned out to be difficult (*Galera et al., 2021*).

The two invasions taking place on Macquarie Island being compared to ''one region/different species'' scheme differ in three factors (factors 10, 15 and 19, Table S1) and only one of them is related to the ''species context'', *i.e.,* 'vegetative propagation'. However, the results of this comparison are uncertain due to the lack of published data.

Our comparison shows that despite King George and Macquarie Island invasions are similar in environmental conditions (polar climate) and the target species (annual plants creating seed banks), they differ in eradication feasibility. Out of the two invasions on Macquarie Island the *S. media* invasion is better documented than *P. annua*, therefore its eradication feasibility can be more easily estimated. This shows that the usage of our system makes it easy to reveal the overlooked aspects of invasion. On the other side of the spectrum, a proper assessment of a very well researched invasion at King George Island would be an important tool to start a definite eradication plan and actions.

## Benefits of USEF usage

USEF offers a simple tool for general and easy eradication assessment and comparison, by covering and organizing a broad range of terminology. Our system is intended to be used by researchers on site, to take under consideration as many factors as possible into eradication preparations and help them share the data further in a universal and accessible manner. Eradication prioritization on islands is a good example where our system can be used. Islands scattered among different climatic zones can be exposed to vastly differing conditions and therefore comparing locally devised frameworks can be difficult and not informative (*e.g.*, *Dawson et al., 2014*; *Helmstedt et al., 2016*; *Stanbury et al., 2017*).

Defining a universal scoring system for eradication associated factors is challenging. So far published papers using prospective scoring systems, simple scales or categorization for evaluating eradication impedance considered local specificity of the eradication course, but lacked universality (see *e.g.*, *Cunningham et al., 2004*; *Panetta & Timmins, 2004*; *Blood et al., 2019*). We propose a simple ordinal evaluation scale, which can be easily amended, altered

or adapted to the local conditions, or reported next to other scales used in ongoing actions. In our example (Table S1) we attempted an eradication feasibility scoring that would not relay on relativity to other compared invasions. The developed universal scoring system of eradication associated factors proposed by us should be reevaluated after accumulation of experience reported in accordance with the unified description system.

## CONCLUSIONS

We believe that the time is ripe for the attempt to organize and summarize the system of assessing the feasibility of successful eradication as well as unifying existing terminology. We propose a detailed clarification of used terms and a simple evaluation scale for the benefit of communication between researchers involved in eradication of invasive species worldwide. The use of a common terminology and scoring system may help to synthesize and draw conclusions from different eradications in different systems, environments and parts of the globe governed by different policies.

Definitions in USEF are broad and of general nature, so the system can be used for different locations across the globe, even unique ones in extreme climate. This is particularly important as the factor highlighted in one framework based on one species/location may not be as important in a different target location. The system is also not restricted to the factors that we now specified. It is open and expandable, which makes it easy to be altered for a specific location needs and further development. Addition of new factors, components or aspects will probably draw the necessity to restructure some definitions to make it even more universal and useful.

The analysis of commonly accepted factors influencing the success of eradication may be useful for:

- creating the strategy for controlling invasive alien plant species—to serve as a "startup checklist" before starting an eradication campaign and to check if a specific infestation eradication may be successful and what are the main factors hindering eradication (*Williams et al., 2019*),
- evaluating the progress of ongoing eradications with time,
- comparison of eradication effects of invasive populations of the same species from different geographic regions during ongoing eradication campaigns,
- comparison of eradication feasibility or efficiency of different invasive species,
- prioritizing "sleeper weeds" for eradication—selection of alien species to be removed first from a given region (*Cunningham et al., 2004*; *Branquart et al., 2016*).

### Funding

This study was carried out at the Biological and Chemical Research Centre, University of Warsaw, established within a project co-financed by the EU European Regional Development Fund under the Innovative Economy Operational Program, 2007–2013.

The funders had no role in study design, data collection and analysis, decision to publish, or preparation of the manuscript.

### Grant Disclosures
The following grant information was disclosed by the authors:
Biological and Chemical Research Centre, University of Warsaw.
EU European Regional Development Fund under the Innovative Economy Operational Program, 2007–2013.

### Competing Interests
The authors declare there are no competing interests.

### Author Contributions
- Halina Galera conceived and designed the experiments, performed the experiments, analyzed the data, prepared figures and/or tables, authored or reviewed drafts of the paper, and approved the final draft.
- Agnieszka Rudak performed the experiments, analyzed the data, authored or reviewed drafts of the paper, and approved the final draft.
- Maciej Wódkiewicz conceived and designed the experiments, authored or reviewed drafts of the paper, and approved the final draft.

### Data Availability
 This work is a literature review and has no raw data or code.

### Supplemental Information
Supplemental information for this article can be found online at http://dx.doi.org/10.7717/peerj.13027#supplemental-information.

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
