# Peer review of "Unified system describing factors related to the eradication of an alien plant species"

_PeerJ, doi:10.7717/peerj.13027_

## Round 0.1 · original submission · Major Revisions

Dear Authors,
Your work has been reviewed by two experts. They both agreed that the work could be published in PeerJ, but had to be thoroughly revised beforehand. Below I am sending comments from both reviewers. Please kindly read them carefully and respond to them.
With best regards,

Reviewer 1 ·

Basic reporting

Regarding basic reporting, I feel the article needs revision before it can be published.

The English writing needs to be improved to improve readability and reduce ambiguity.

I think the article will need to be reframed. I feel that standardising the terms used for factors that may influence eradication should not be the primary focus. What is more important is developing a standardised approach to scoring eradication success that this easy to use. I think the authors make good progress with this, but more specificity is also needed for each variable to allow for standardisation and comparison between sites.

I think the framing and objectives of this work need to be more clearly thought out and stated. (I think standardising terminology, should not be a core objective, but rather standardising socring).

The authors need to should how their work builds on previous studies e.g. Dodd, Gardnerner (a suggested new reference)

Experimental design

I think the review of factors is good. A little more is needed to show how the author's systems build on that of previous work especially Dodd et al.

I feel the new approach needs to be tested for more eradication projects globally. I think this will help to develop methods and metrics to improve standardisation to allow for comparison globally.

I think that for each factor standardisation is needed, and different catalogers to choose from should be offered for each one. Linking to this a confidence score should also be added for each.

Validity of the findings

I think the findings and introduction and conclusion do not link so well and there is a need for some reframing (more about standardised scoring). Based on this I feel the authors should be reading and getting inspiration from other scoring and reporting frameworks (not just those based on eradication). Some examples include:

Volery, L., Bacher, S., Blackburn, T.M., Bertolino, S., Evans, T., Genovesi, P., Kumschick, S., Roy, H.E. and Smith, K.G., 2020. Improving the Environmental Impact Classification for Alien Taxa (EICAT): a summary of revisions to the framework and guidelines.

Bacher, S., Blackburn, T.M., Essl, F., Genovesi, P., Heikkilä, J., Jeschke, J.M., Jones, G., Keller, R., Kenis, M., Kueffer, C. and Martinou, A.F., 2018. Socio‐economic impact classification of alien taxa (SEICAT). Methods in Ecology and Evolution, 9(1), pp.159-168.

Blackburn, T.M., Essl, F., Evans, T., Hulme, P.E., Jeschke, J.M., Kühn, I., Kumschick, S., Marková, Z., Mrugała, A., Nentwig, W. and Pergl, J., 2014. A unified classification of alien species based on the magnitude of their environmental impacts. PLoS biology, 12(5), p.e1001850.

Wilson, J.R., Faulkner, K.T., Rahlao, S.J., Richardson, D.M., Zengeya, T.A. and Van Wilgen, B.W., 2018. Indicators for monitoring biological invasions at a national level. Journal of Applied Ecology, 55(6), pp.2612-2620.

Pheloung, P.C., Williams, P.A. and Halloy, S.R., 1999. A weed risk assessment model for use as a biosecurity tool evaluating plant introductions. Journal of environmental management, 57(4), pp.239-251.

Shackleton, R.T., Bertzky, B., Wood, L.E., Bunbury, N., Jäger, H., van Merm, R., Sevilla, C., Smith, K., Wilson, J.R., Witt, A.B. and Richardson, D.M., 2020. Biological invasions in World Heritage Sites: current status and a proposed monitoring and reporting framework. Biodiversity and Conservation, 29(11), pp.3327-3347.

Additional comments

I have reviewed the paper looking at a unified system for describing factors related to the deracination of plant species. I think the paper has merit but I have some reservations at the moment and believe work is needed to improve the manuscript. I think the study is not so well framed and makes it a bit hard to follow and see the value of the work. I also feel it needs to be standardized more (a choice of categories for each factor, and better tested using more case studies globally).

Firstly, you make an argument for a unified system of terminology; however, I am not if there is any controversy or many issues with the use of the terms included in table one. Yes, people call certain things differently which is ok, there are just variables for analysis of the potential success of an eradication campaign. I am certain the lack of standardization is not causing any major problems. There is more contention over core definitions like the word “eradication” itself that is an issue. I am not sure “unifying” the names of factors that could affect eradications is that necessary or important. Previous “unifying” studies have tried to merge different frameworks or standardize key terms that are used massively like “invasion” “eradication”, “naturalization” “management”, “pathway” etc. I would rather frame this article as a review and consolidation of factors that influence the eradication feasibility of plants. I would also go further and make it a detailed scoring tool see below). I think this is where the interest will be and less so on terminology standardization. Linking to this comment and list of variables, the authors cite Dodd et al. (2015) who identify 22 factors. Their final set of variables proposed here is 24. Based on this I am not sure how much this work actually adds in this regard either. The needs to be elaborated on and your contributions to science need to be better highlighted. Linking to this as well in your article you choose not to define the values or categories for the factors, however, to collect comparable data this is essential and I believe that through the literature review you did you could come up with meaningful categories. If this is not done, the data collected using the factors you propose will be highly subjective and incomparable.

Secondly, the English writing needs improvement, I know the authors are not first language English speakers and I know how difficult it must be to write in a “foreign” language but the current manuscript is hard to understand at times and the authors need to improve it. I do highlight some key changes needed by there are many more minor ones.

Thirdly, I also feel that the new set of proposed standardized variables to assess eradication success or failure should be tested on more than just three examples. This is not enough to test how robust the system is. I believe that many more case study examples should be tested and compared in your study. I would even go as far as testing 30 or more and running some statistical analyses on the data. Maybe focusing on a region (artic islands, or Europe etc.) could work (e.g. see the study you cite by Pluess et al 2012). The best thing is if your list could be used to predict the success an eradication campaign might have.

Finally, I am still a bit sceptical about this framework and its lack of specificity. I think it needs more testing. I also think the authors should look at similar assessment or scoring tools like weed risk assents and impact scoring tools to guide and improve their work.

Minor comments

Line 25-27 and Figure 1: I think calling it “site context” instead of “location context” would be best. Similarly, “social” context instead of human context would be better. Again “replay” is not the correct word to use. How about “reinvasion”, I do see later you discuss why you call it “replay” but it’s a bit colloquial and I am not sure if this is the correct word to use. Maybe “reassessment” would work but without “context” afterwards.

Line 25-25: This is hard to understand.

Line 28: I am not sure what is meant here.

Line 39-43: I would remove the quote, I don’t think it is really relevant for this paper.

Line 45. Delete

Line 46: “Biological invasions” is not a field, it is a process or phenomenon. Invasion science or invasion biology is the field. This needs to be changed.

Line 47: Reword this sentence. In rewording this sentence also change “prevention and combat” to “management”.

Line 58 -: I don’t think terminology is such an issue. It is more about identifying factors that contribute to or hinder the success of eradication. There are major terminology debates around what eradication means, but I am not sure if this fits in the scope of this paper. I think you should argue more that it is important to identify a standardized set of factors to analyze eradications, not to standardize the use of terms for these.

Line 67-73: This section is much better and focused on the study. I would personally start your introduction here, and cut out most of what was written above. Small parts of it can be incorporated below.

Line 77: I think a key paper that is missing is: Gardener, M.R., Atkinson, R. and Rentería, J.L., 2010. Eradications and people: lessons from the plant eradication program in Galapagos. Restoration Ecology, 18, pp.20-29.

Line 81: The name of your system does not make so much sense. I would rather call it “a system (or framework) for assessing eradication feasibility”.

Line 85-87: I have said this many times now, but to me, standardization of these terms are not important rather what is important is identifying and testing a full suite of factors that affect eradication campaign successes. I think the paper should be framed in this manner.
Line 89-90: I am not so sure what this means.

Line 91: “is now more …. to globalization” – delete this.

Lines 92-94: A little repetitive now.

Line 97: change “check” to “search”

Line 103: Please provide the citations for all the papers reviewed in a supplementary material file.
Methods: Please list all the exact details that were extracted from each of the reviewed papers. How did you go about collecting and compiling the data from the papers?

Line 149: I disagree with the authors here. The “number and distribution of propagules” relates to the size of the invasion – the associated term you put here relate to dispersal traits. They are not really comparable or the same in my mind. Maybe if it was termed number and dispersal it would be ok?

Line 176-179: I disagree, if you are doing it on a global level this needs to be done as precisely as possible otherwise you will be collecting different data based on people’s different subjectivities. Even with a high level of standardization personal understanding, subjectivities will still be an issue. I think you need to develop clear guidelines for this and things need to be as standardized as possible. This can include ranges. Similar approaches like EICAT that analyze and compare impacts of various invasive species in different systems globally. Without making standardizing values this approach will end up comparing apples with oranges. Based on the literate you should be able to identify some ranges to categories values e.g. infestation size - < 1 m2 ; 1 – 10 m2 ; 10 – 50 m2 ; 50 – 100 m2 ; 100 – 500 m2 ; 500- 1km2 ; 1 – 5 km2 ; 5-10 km2 ; 10-50 km2 >50 km2 . I think setting broad scales that account for differences in social and ecological systems is needed. Land uses, could be protected, natural, semi-natural, agricultural, urban, etc.

Line 196: For eradication, there needs to be an elimination of the total population (0 for at least a certain length of time) otherwise if it is just lowering it, that would be classified as containment or asset protection, not eradication.

Line 274: This might need to be broader than just climate? Soils, fire regimes etc.?

Line 288:Needs clarification.

Line 295: I would stick with “reinvasion”.

Line 301. Is it not more for scientists who would collect and analyze available information?

Line 304: I would not say this, you put doubt in the reader's mind about what your work contributes.

Line 305: Not without some form of standardizations for each factor.

Line 320-325: Wouldn’t this be best to identify using statistical approaches. You could assess the importance of factors within a given context using multiple regression? Again, this scoring system lacks guidelines and so maybe too subjective.

Lin 336: I think it needs clearer guidelines.

Line 530 onwards: As mentioned I think you should use a lot more cases to test this to see how robust and standardized and useful the proposed set of factors can be. Alternatively, it can be limited to fewer case studies but it might be useful to get multiple people to score it to see if there are similarities or differences in how it is applied based on different assessors. To check how robust the scoring is it might be useful to add a confidence score for each of the factors.

Figure 1:
Point 6: “land use” and “ownership” should be separated into two different variables. They are not mutually exclusive.
Pint 8: What about adaptation to other things: should this be adaptations to the local environmental context, not just climate change.
I think there should be some predefined categories to choose from for each of the 24 factors.

Reviewer 2 ·

Basic reporting

This paper proposes a simple system describing factors related to the eradication of an alien species. This is an interesting paper. It is well written, well-structured and brings new information.

Experimental design

(a) One of the problem is that some the factors are of categorical nature, while some other are continuous. In this ms, all factors are treated as categorical. Does this not reduce the power of the method?
(b) The last context, “replay context”, which contains only one factor, “reinvasion possibility (factor 24)”, should not be produced mathematically by the previous ones?
(c) The authors should make it clear that this method can be applied at a local scale. For example, this cannot be applied at a national scale, where the same factor (e.g. monitoring area size) may be labeled differently in one location (an island population) than another for the same species (a mainland population).

Validity of the findings

no comment

Additional comments

(d) I think that the Table S1 should be part of the main text.

Minor comment
(e) Survey methodology: authors have to give the time-period of publication of papers they searched for. In addition, we need a criterion for the final selection of 32 papers.

---

## Round 0.2 · accepted · Accept

Noble Authors,
After a deep analysis of the corrections that you led to the work, I believe that it may be published in the current version.
My sincerest congratulations!